# Variability and the Correlation of Kinematic and Temporal Parameters in Different Modalities of the Reverse Punch Measured by Sensors

Vesna Vuković [1,*], Anton Umek [2], Milivoj Dopsaj [1], Anton Kos [2], Stefan Marković [1] and Nenad Koropanovski [3]

1 Department of Physical Education and Health, Faculty of Sport and Physical Education, University of Belgrade, 11000 Belgrade, Serbia; milivoj.dopsaj@fsfv.bg.ac.rs (M.D.); stephan.markovic@hotmail.com (S.M.)
2 Department of Information and Communication Technology, Faculty of Electrical Engineering, University of Ljubljana, 1000 Ljubljana, Slovenia; anton.kos@fe.uni-lj.si (A.K.)
3 Department of Criminalistics, University of Criminal Investigation and Police Studies, 11800 Belgrade, Serbia; nenad.koropanovski@kpu.edu.rs
* Correspondence: vbvukovic@gmail.com

**Abstract:** The influence of joint motion on punch efficiency before impact is still understudied. The same applies to the relationship between the kinematic and temporal parameters of a reverse punch (RP) that determines a score. Therefore, the aim of this study was to investigate if the exclusion or inclusion of body segments affects the acceleration, velocity, rotation angle, and timeline of execution, and to examine the correlation between these quantities. Seven elite male competitors—senior European and World Championship medalists—participated in the in-field testing. Quantities were acquired in the developmental phase of RP through three modalities of execution. Synchronized real-time data were obtained using combined multimodal sensors and camera fusion. The main findings of the study have highlighted the significant differences in the temporal and kinematic variables of RP that arise from the modality of execution. Large and medium correlation coefficients were obtained between the examined variables of body and hand. In conclusion, the results show that measured parameters are affected by segmental body activation. Moreover, their interdependence influences punch execution. The presented interdisciplinary approach provides insightful feedback for: (i) development of reliable and easy-to-use technical solutions in combat sports monitoring; and (ii) improvements in karate training.

**Keywords:** karate; technique analysis; acceleration; velocity; rotation angle; timeline; wearable devices





## 1. Introduction

The difference between a punch that is point-worthy and one that is not can be a few milliseconds. Besides the permanent adjustment imposed by the karate combat environment [1] and a number of adaptive interactions between the nervous and musculoskeletal systems [2], the relationship between the kinematic and temporal parameters of the motion may be one of the main determinants of a successful outcome. Strictly speaking, attention should be directed to acceleration, velocity, rotation angle, and temporal parameters. The temporal parameters are defined as the equivalent of the kinematic event occurring during the corresponding characteristic phase of the punch execution in the optimal time sequence. Joint kinematics are traditionally restricted to laboratory research [3,4]. Available detection systems are typically robust, expensive, and demand trained experts who know how to operate them [5,6]. In such settings, athletes face limitations that are not common in competition and training [6]. Consequently, constraints imposed by the equipment and environment may affect the motor skill under analysis [3]. In contrast, the acquisition of a wide range of diverse data in situ using kinematic sensors (KS) enables measuring during

real-time activities. A method like this produces objective qualitative and quantitative analyses, as well as enhanced motion estimation [7–10]. A better understanding of the kinematic scheme of a punch would lead to technical improvement and the practices of top athletes shared among throwing and striking sports [11], such as karate.

The fight-oriented style of contemporary karate retains the essence of traditional martial arts, but it has evolved into a complex sport with two distinguished disciplines, one of which is combat [12]. Karate combat is characterized by alternating periods of high and low intensity, short, ballistic offensive and defensive actions, and great variability [13–15], with the final intention of outperforming the opponent with a more efficient technique [16]. Karate competition can take the form of individual or team competition in a match lasting three minutes [16]. The majority of time is constituted of aerobic activity [13,17], a great deal of which includes dynamic movements, preparation for offensive actions, and adaptation for defense [15,18]. In short periods of anaerobic alactic activity, competitors execute pointing actions. In order for the actions to be scored, the technique has to meet quality standards: action, whether offensive or defensive, has to be applied with a high level of energy, speed, and good timing—when its potential efficiency is the greatest [16]. The same standard applies to a kick or a punch.

Punching techniques are essential in many combat sports [19], karate being one of them. Compared to kicks, punches are significantly more common in elite karate competitions, accounting for 89.09% of the total frequency. A considerable number of points are due to the reverse punch—RP. RP accounts for 66.91% of the points, with more than half of them attained by aiming at the body [18]. It is a demanding task to perform an efficient punch. To achieve this, optimal involvement of all biological abilities is needed in the restricted time [20]. Previous research has mainly focused on the final phase of the RP, i.e., the impact. Results show that the higher the execution speed, the greater the energy and power transferred to the opponent [1]. In addition, it was reported that the timely inclusion of body parts in the sequential structure of the movement affects punching speed [8,9], but the relationship between the kinematic and temporal parameters stayed vague. The latest research revealed that RP at different speeds has a recognizable temporal structure. The study showed that in the total movement timeline, the fastest punch is always characterized by earlier hand movement, and the maximal velocity of the hand reached closer to the impact [9]. Several phases are recognized in punch execution, as well as different speeds of contralateral upper limbs during performance [21]. It has been documented that kinematic and neuromuscular activity in the punch occurs within 400 ms, with a proximal to distal sequence of activation [22]. The peak of the wrist linear velocity reaches up to $7.3 \pm 0.8$ m/s in elite athletes [23]. It is found that athletes' competing experience makes a difference in perceptual and reaction skills, neuromuscular control indicators, and performance. The elite karate athletes react and accelerate the wrist more quickly than sub-elites. In addition, they are able to brake more quickly, which makes the technique less obvious to the opponent [24]. Saponara [25] found that in the first phase of the movement lasting 160 ms, hand speed increases continuously from 0 to 12.5 m/s. The average acceleration obtained in this research was about 78 m/s$^2$. Considering the contribution of the lower part of the body in the punch execution [19], this should also be taken into account. That is, through rotation of the pelvis, or the body, the joint sequence of the lower limb is linked to the joint sequence of the arm [26]. It is found that a greater shoulder extension is a characteristic of karate fighters. They are also able to generate high values of preferred velocity to perform joint movements [27]. Studies revealed that different task orientations and conditions of execution influence punch kinematics. For example, acceleration values differ depending on the primary orientation towards force or speed at a fixed or self-selected distance [1]. Moreover, changing distance length has a significant influence on force production and punch acceleration [28]. For these reasons, the analysis of RP should be founded on the conditions that replicate training and combat environments. Contrary to that, a laboratory approach is dominant.

Lately, researchers have offered various systems tending to improve combat sports training. They rely on technological advances (computer vision pose estimation, vision and inertial sensing system, sensors, and convolution neural networks, novel non-conventional system), aiming to understand punch execution and develop efficient tools for monitoring and enhancing combat technique in different aspects [21,29–31]. However, joint kinematics is traditionally restricted to laboratory research [3,4]. A frequently employed method in the biomechanical analysis of punching techniques in studies, in addition to the force platform [20,32] and electromyography [22,33], is motion capture carried out with marker-based optical technology (sometimes combined with force platforms) [20,23,34–40]. Unlike other proposed systems, force platforms are not able to provide information about the movement prior to contact, which disables insight into the developmental phase of the punch. EMG overcomes this problem, but its main disadvantage is that the apparatus is attached to the body, which limits the athlete's movements. Apart from that, performing punches in dynamic conditions is a special challenge because of the complexity of movement in real settings and possible noise due to speed or muscle size. Physiological processes that are more pronounced or only occur under conditions of high exertion might also affect the shape or size of the corresponding EMG signal. In addition, electromyograms must be normalized in order to facilitate interpretation, offer adequate reliability, and provide a representative measure of muscle activity [41]. The marker-based optical systems are highly reliable at providing spatial location data, but they suffer from shortcomings when it comes to the acquisition of kinematic quantities [3]. Above all, the application of such equipment requires controlled conditions, numerous markers, and cameras, as well as the involvement of complex algorithms for feedback [40]. On the other hand, the dynamic sports performance environment imposes the adaptability of complex motor patterns. Athletes are forced to choose their movement strategies through a self-organization process in order to discover a proper response within these limits [42]. This condition is compromised in a controlled environment, such as a laboratory [6]. The application of KS technology in typical combat situations could be the proper solution for measurement in real surroundings, enabling data acquisition in an accurate and non-intrusive way [5,6,10]. With a setup that is simple to use without any sophisticated arrangements, wearable sensors are user-friendly [43]. The ballistic motion of limb segments, as well as rapid intervals of acceleration and deceleration, such as punching [44], should be the starting point in choosing a device with appropriate measuring characteristics.

As already addressed, in conventional research, athletes face limitations that are not common in realistic sports conditions. Consequently, constraints imposed by the equipment and environment may affect the motor skill under analysis [3]. Knowing that top-level sport deals with fine modifications and responses to variations, the implications are obvious: Data acquired are biased by measurement settings. One more problem occurs: The diversity of the methodological approach in the small number of existing studies makes it difficult to compare and systematically present the kinematic findings in order to obtain a comprehensive biomechanical illustration of the analyzed motor task. As well as the previously mentioned problems, the main limitations of the referred to studies are: (i) the diversity of participants and comparison conditions, (ii) the various test conditions which influenced the monitored kinematic parameters, (iii) the different phases of punch execution or kinematics of particular segments under analysis, and (iv) the different positioning of the apparatus, marking and tracing the reference points. Apart from that, it is well known how the pointing technique should be in terms of kinematics at impact. However, how quantities of interest are developing in order to result in a successful punch is not clear yet. This applies equally to variability and correlation of kinematics and temporal parameters. Therefore, the aim of the current study is movement sequence analysis, which could lead to a better understanding of RP aimed at the body. We hypothesized that (i) the kinematics of RP in the developmental phase change due to technique modality and (ii) there is a correlation between the kinematic and temporal parameters of the body and hand in the developmental phase of punch.

The major scientific contributions of the study are: (i) a novel, simple-to-use methodology employing specific test and kinematic sensors for measuring the temporal and kinematic quantities of interest for successful punch execution, (ii) revealing the structure of the punch in the developmental phase in RP, (iii) providing insight into relations between the parameters relevant for technique performance, and (iv) providing a valuable base of knowledge for the training of elite athletes.

## 2. Materials and Methods

The quasi-experimental approach was chosen in this study in order to explore real-sport situations.

### 2.1. Participants

Seven elite male competitors, members of the Serbian national team—senior European and World Championship medalists with valid medical certificates—voluntarily participated in the in-field testing. The study was conducted immediately upon the completion of the National Cup, one of the two major competitions of the season, implying they were all at a high level of performance. The basic descriptive characteristics for the sample were as follows (mean value $\pm$ SD): age—$20.63 \pm 2.07$ years; height—$1.86 \pm 0.04$ m; body mass—$82.25 \pm 6.69$ kg; experience—$10.50 \pm 1.60$ years. The study was conducted following the ethical standards recognized by the Declaration of Helsinki and was approved by the Ethics Research Committee.

### 2.2. Procedure

Participants completed a 15-min standardized warm-up session that included general exercises, followed by specific combat punching techniques in various modalities, which, from the first to the last, differed in technical complexity. The technical complexity is reflected in the inclusion of body parts that support the punch, which gradually changes the conditions of the punch execution from static to dynamic. This approach replicates the karate training system. Commonly, novice training starts with *choko-tsuki* (ChT) [22]. ChT is a form of RP practiced to bring the practitioner's focus to the path of execution, limb coordination and timed muscle contraction. The next task in the karate curriculum is mastering hip movement control, often practiced in a *zenkotsu-dachi* (basic stance) [45]. After this stage, more demanding tasks are introduced in training, one of which is performing the punch while moving in a way that simulates combat in *fudo-dachi* (combat stance).

Two initial stances were adopted in testing: *zenkutsu-dachi* (Figure 1a,b) and *fudo-dachi* (Figure 1c,d). After they were familiarized with the procedure, the participants performed a test that consisted of RP in three different modalities:

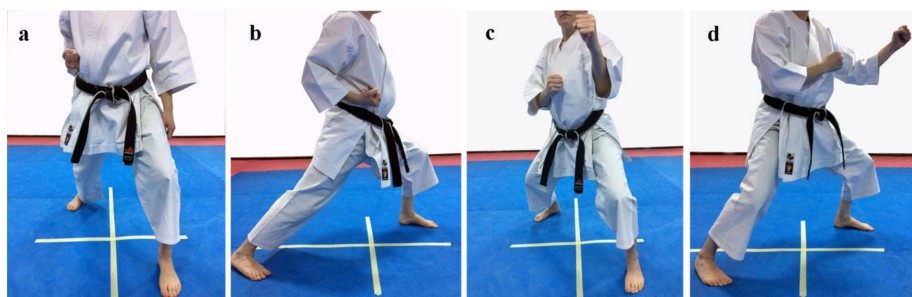

**Figure 1.** Starting positions in three modalities: (**a**,**b**) *zenkutsu-dachi* (basic stance) front and side view; (**c**,**d**) *fudo-dachi* (combat stance) front and side view.

(i)    RPNH: static punch execution with feet firmly on the ground and no hip rotation included. The *zenkutsu-dachi* was the initial stance and required positioning both hips at 90 degrees relative to the stance direction (i.e., front position of hips).

(ii)   RPH: static punch execution with feet firmly on the ground, but hip rotation included. The *fudo-dachi* was the initial stance and required hip positioning at an angle relative to the stance direction (i.e., open position of hips).

(iii)   RPSM: punch execution in motion including hip rotation. Stance and hip position were the same as in the second test. The motion adopted was the common pattern motion combined with the reverse punch. Thereby, the sequence of the test was: static starting position—sliding movement (dynamic preparation stage)—execution (dynamic execution stage) that starts by attaining stability in combat stance again, for a very brief, transitional moment.

Given the fact that all the participants were right-handed, the technique was executed in the front left stance. Approximately five seconds after hearing the signal, participants performed three consecutive punches from a static position. Each time after performing the punch, the participants returned to the starting position, repeating the punch three times in a row in the shortest time. This was regarded as an attempt. The reasoning behind this is that karate combat typically involves time-limited actions characterized by multiple punches [18]. Due to that, the athlete needs to adapt in the shortest amount of time, which was replicated in the test. Participants had two attempts and were given enough time to rest between the attempts and modalities. The attempt was regarded as successful if all three punches met the technical criteria (i.e., the quality of the technique was sufficient for awarding points). It is likely that, in such conditions, some of the punches will not be as effective or fast as they may be. They must nevertheless be taken into account, as they support a fighter's ability to apply an efficient technique while dealing with a time limit. To ensure participants maintained an appropriate level of performance, the testing was monitored by the three high-ranking referees who have officiated at numerous world and continental championships.

### 2.3. Measurement and Data Processing

The primary source of power for a variety of sports, which can give speed or acceleration to performed techniques and manage their magnitudes, are movements of the trunk and limbs [46]. Due to that, the kinematic characteristics of the RP were obtained using two custom-made wireless KS devices. The metrical characteristics and reliability of the device were confirmed [8]. KS were positioned on the hand and the back of the athlete as follows: (i) on the upper side of the fist performing the RP, between the second and fourth metacarpal bones, and (ii) in between the second and third lumbar vertebra, respectively. The positioning was based on the kinetic chain of technique and factors, excluding any source of disruption for KS or an athlete, as well as providing the most valuable data [26]. The device attached to the hand consisted of a 6DoF sensor LSM6DS33, a 3D accelerometer, a 3D gyroscope, and a microcontroller with a communication WiFi module. The accelerometer's dynamic range was $+/-$ 16 g, and the dynamic range of the gyroscope was $+/-$ 2000 dps. The acceleration signal was captured with a sampling frequency of 200 Hz and transmitted to a personal computer via a WiFi module using the UDP protocol. A particular phase of RP preceding the impact is characterized by a progressive increase in kinematic quantities, making the operating range of the devices and sampling frequency sufficient. The device attached to the location corresponding with the center of gravity on the athlete's back operated in the same way as the one on the hand but used a different type of sensor. It had a built-in absolute orientation sensor BNO055 and measured linear acceleration at a lower sampling frequency of 100 Hz.

In order to provide additional check-ups during post-processing, two cameras were used to record the testing: the GoPro HERO 6; GoPro Inc., San Mateo, CA, USA (frequency 100 fps, resolution 1920 × 1080 p) and the Logitech C920 HD PRO; Logitech Inc., Lausanne, Switzerland (frequency 15 fps, resolution 1280 × 820 p). Cameras were placed on a 1.3 m tripod, 2 m laterally (on both sides) in relation to the athlete. Such positioning was considered adequate because it met two conditions: (i) enabling enough space for uninterrupted task performance in the field of view and (ii) recording punch performance

from start to end point. Additional check-ups were possible on account of combining images from different viewpoints and sensor signals, allowing movement phase recognition and differentiation [47,48]. Heterogeneous but temporally synchronized data (Figure 2) acquired from four devices decrease uncertainty in technique evaluation. Data were stored in video (.avi and .MP4) and LabView (.tdms) files. Two wireless sensors sent signals to the LabView program. The signals from the sensors and the two cameras were synchronized and recorded. Post-processing and further analysis were carried out with MathCAD 7 numerical calculation software. A Butterworth 5th-order low-pass filter with a cut-off frequency of 40 Hz was used to analyze the signal. Shifting the signals at the reference moment of analysis—an impact—was the initial step in calculating the variables of interest [9,26].

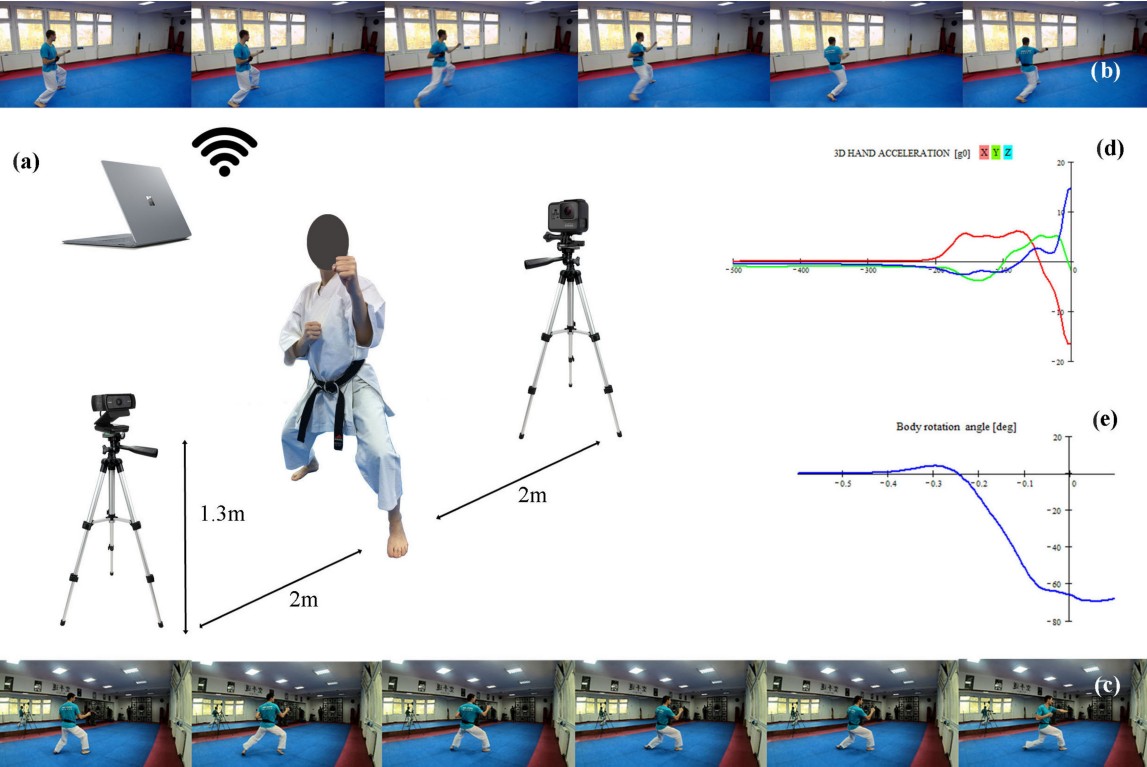

**Figure 2.** Experimental setup (**a**) for karate testing and an example of the multimodality data consisting of two cameras—GoPro HERO 6 (**b**) and Logitech C920 HD PRO (**c**) video files, and two sets of internal data acquired from the sensors positioned on the hand (**d**) and body (**e**).

### 2.4. Variables

As acceleration, velocity, and body rotation were marked as key kinematic components of a punch, they were chosen as variables of interest. The maximum values in the phase of the punch development were measured, and the recorded time of the onset of a movement and the peak of the kinematic variable on the primary axis of motion were taken into account.

The temporal variables were:

- tHAS—time for the onset of hand acceleration;
- tHA—time for the maximum hand acceleration;
- tHV—time for the maximum hand velocity;
- tBAS—time for the onset of body acceleration;
- tBA—time for the onset of body acceleration;
- tBV—time for the maximum body velocity;
- tBRa—time for maximal body rotation angle.

The kinematic variables were:

- HA—maximum hand acceleration, expressed in g0;
- HV—maximum hand velocity, expressed in m/s;
- BA—maximum body acceleration, expressed in g0;
- BV—maximum body velocity, expressed in m/s.
- BRa—maximal body rotation angle, expressed in deg.

BRa, HA, and BA were primary variables, while HV and BV were derived as integral to the acceleration of the hand and body. The calculations were based on the model of one-dimensional movement.

### 2.5. Statistical Analysis

In the first processing step, mean and standard deviation were calculated for the raw data. The normality assumption was tested using the Shapiro–Wilk Goodness of Fit test. The nonparametric Kruskal–Wallis test was adopted to examine the general differences in the temporal and kinematic parameters between the three modalities of RP. In the next step of the analysis, pairwise comparisons using the Mann–Whitney U test were conducted, applying the Bonferroni correction. The effect sizes *(r)* were also calculated for a focused comparison [49]. In the final step, partial rank correlation [50], using execution type as a control variable, was employed to establish possible relationships between the temporal and kinematic parameters of RP. The power of correlations was defined according to a study by Cohen [51]. The level of statistical significance was set to the probability of 95%, i.e., $p \leq 0.050$. The statistical analyses were carried out using Microsoft Office Excel 2007 and the Statistical Package for the Social Sciences (IBM, SPSS Statistics, version 20.0) for Windows.

## 3. Results

Descriptive statistics for the raw data (kinematic and temporal parameters) in the three modalities of RP are presented in Table 1.

**Table 1.** MEAN $\pm$ SD calculated for the kinematic and temporal parameters in different modalities of RP.

| Variable/ Test | RPNH | RPH | RPSM |
|---|---|---|---|
| tHAS (ms) | −155.89 ± 28.98 | −207.89 ± 42.97 | −242.38 ± 39.37 |
| tHA (ms) | −89.89 ± 23.44 | −79.44 ± 27.88 | −83.21 ± 32.53 |
| tHV (ms) | −40.56 ± 15.71 | −33.33 ± 13.01 | −32.50 ± 12.01 |
| tBAS (ms) | −138.11 ± 48.21 | −211.33 ± 37.88 | −228.21 ± 53.13 |
| tBA (ms) | −309.89 ± 201.28 | −196.11 ± 25.96 | −254.17 ± 125.06 |
| tBV (ms) | −87.78 ± 64.21 | −44.22 ± 36.49 | −55.00 ± 38.62 |
| tBRa (ms) | −15.22 ± 35.72 | 38.78 ± 39.77 | 29.29 ± 50.64 |
| HA (g0) | 5.92 ± 1.64 | 6.73 ± 1.22 | 6.48 ± 1.05 |
| HV (m/s) | 3.89 ± 0.90 | 6.20 ± 0.70 | 6.58 ± 0.64 |
| BA (g0) | 0.21 ± 0.34 | 0.51 ± 0.25 | 0.31 ± 0.21 |
| BV (m/s) | 0.10 ± 0.10 | 0.97 ± 0.41 | 1.18 ± 0.20 |
| BRa (deg) | 14.35 ± 4.96 | 74.95 ± 15.02 | 64.76 ± 13.93 |

Abbreviations: RPNH—reverse punch no hip included; RPH—reverse punch hip included; RPSM—reverse punch in motion; t—time of event; H—hand; B—body; A—acceleration in the dominant axis of motion; V—velocity; Ra—rotation angle; S—start.

The Kruskal–Wallis test revealed significant general differences across all three modalities in all the examined variables but tBA (Table 2).

**Table 2.** The general differences in the temporal and kinematic parameters of the reverse punch examined across the three modalities of RP.

|  | tHAS | tHA | tHV | tBAS | tBA | tBV |
|---|---|---|---|---|---|---|
| Chi–Square | 73.64 | 9.44 | 6.51 | 57.60 | 5.78 | 20.04 |
| df | 2 | 2 | 2 | 2 | 2 | 2 |
| Asymptotic Significance | 0.000 | 0.009 | 0.039 | 0.000 | 0.056 | 0.000 |
|  | **tBRa** | **HA** | **HV** | **BA** | **BV** | **BRa** |
| Chi–Square | 38.91 | 12.49 | 87.21 | 37.59 | 95.02 | 91.12 |
| df | 2 | 2 | 2 | 2 | 2 | 2 |
| Asymptotic Significance | 0.000 | 0.002 | 0.000 | 0.000 | 0.000 | 0.000 |

Abbreviations: t—time of an event; H—hand; B—body; A—acceleration in the dominant axis of motion; V—velocity; Ra—rotation angle; S—start.

Further analysis identified partial differences. It can be seen from the data in Table 3 that an increase in HV, BV, BA, and BRa was significantly different ($p < 0.050$) across the three modalities of execution, with the lowest effect size between RPH and RPSM. As for the HA, significance was not proven between RPH and RPSM alone. Statistical significance was not proven for five temporal variables between RPH and RPSM.

**Table 3.** The pairwise comparison and the effect size of the temporal and kinematic parameters of RP in relation to the modality of execution.

| Test |  | tHAS (ms) | tHA (ms) | tHV (ms) | tBAS (ms) | tBA (ms) | tBV (ms) |
|---|---|---|---|---|---|---|---|
| RPNH | U | 290.00 | 677.00 | 759.50 | 231.00 | 827.00 | 511.00 |
| vs. | Sig. | 0.000 | 0.007 | 0.040 | 0.000 | 0.134 | 0.000 |
| RPH | r | −0.615 | −0.286 | −0.217 | −0.665 | −0.158 | −0.427 |
| RPNH | U | 22.00 | 643.50 | 673.50 | 161.00 | 911.50 | 539.50 |
| vs. | Sig. | 0.000 | 0.010 | 0.020 | 0.000 | 0.776 | 0.001 |
| RPSM | r | −0.841 | −0.276 | −0.249 | −0.714 | −0.031 | −0.370 |
| RPH | U | 478.50 | 919.00 | 912.00 | 789.50 | 632.00 | 794.50 |
| vs. | Sig. | 0.000 | 0.824 | 0.777 | 0.186 | 0.008 | 0.201 |
| RPSM | r | −0.425 | −0.024 | −0.030 | −0.142 | −0.286 | −0.137 |
|  |  | **tBRa (ms)** | **HA (g0)** | **HV (m/s)** | **BA (g0)** | **BV (m/s)** | **BRa (deg)** |
| RPNH | U | 265.00 | 622.00 | 40.00 | 328.00 | 1.00 | 0.00 |
| vs. | Sig. | 0.000 | 0.002 | 0.000 | 0.000 | 0.000 | 0.000 |
| RPH | r | −0.637 | −0.332 | −0.827 | −0.582 | −0.861 | −0.861 |
| RPNH | U | 440.50 | 616.00 | 6.00 | 542.00 | 0.00 | 0.00 |
| vs. | Sig. | 0.000 | 0.005 | 0.000 | 0.001 | 0.000 | 0.000 |
| RPSM | r | −0.461 | −0.300 | −0.855 | −0.367 | −0.861 | −0.861 |
| RPH | U | 754.00 | 830.00 | 652.00 | 494.00 | 482.00 | 646.00 |
| vs. | Sig. | 0.104 | 0.329 | 0.013 | 0.000 | 0.000 | 0.011 |
| RPSM | r | −0.174 | −0.105 | −0.267 | −0.411 | −0.422 | −0.272 |

Abbreviations: RPNH—reverse punch no hip included; RPH—reverse punch hip included; RPSM—reverse punch in motion; t—time of an event; H—hand; B—body; A—acceleration in the dominant axis of motion; V—velocity; Ra—rotation angle; S—start.

In the final step, partial rank correlation was applied, and the results of the analysis are presented in Table 4. A large correlation was found between tHV:tHA, BA:tBA, tBV:tBA, BRa:BV, explaining 57%, 36%, 34%, and 30% of the variance, respectively. The medium correlation coefficients were determined for 17 pairs of variables.

**Table 4.** Results of partial rank correlations between the temporal and kinematic body and hand parameters of RP (N = 129).

|  |  | 1 | 2 | 3 | 4 | 5 | 6 | 7 | 8 | 9 | 10 | 11 | 12 |
|---|---|---|---|---|---|---|---|---|---|---|---|---|---|
| 1. | tHAS | – |  |  |  |  |  |  |  |  |  |  |  |
| 2. | tHA | 0.33 ** | – |  |  |  |  |  |  |  |  |  |  |
| 3. | tHV | 0.38 ** | 0.76 ** | – |  |  |  |  |  |  |  |  |  |
| 4. | tBAS | 0.46 ** | 0.29 ** | 0.38 ** | – |  |  |  |  |  |  |  |  |
| 5. | tBA | 0.18 * | 0.26 ** | 0.32 ** | 0.12 | – |  |  |  |  |  |  |  |
| 6. | tBV | 0.01 | 0.25 ** | 0.11 | 0.10 | 0.59 ** | – |  |  |  |  |  |  |
| 7. | tBRa | 0.10 | 0.13 | 0.09 | −0.14 | 0.22 ** | 0.06 | – |  |  |  |  |  |
| 8. | HA | 0.31 ** | 0.33 ** | 0.25 ** | 0.11 | 0.46 ** | 0.18 * | 0.24 ** | – |  |  |  |  |
| 9. | HV | −0.24 ** | 0.27 ** | 0.23 ** | −0.07 | 0.17 | 0.10 | −0.07 | 0.39 ** | – |  |  |  |
| 10. | BA | −0.05 | 0.18 * | 0.19 * | −0.38 ** | 0.60 ** | 0.44 ** | 0.33 ** | 0.39 ** | 0.13 | – |  |  |
| 11. | BV | −0.20 * | 0.08 | −0.02 | −0.24 ** | 0.08 | 0.38 ** | 0.03 | 0.03 | 0.24 ** | 0.25 ** | – |  |
| 12. | BRa | −0.28 ** | 0.19 * | 0.09 | −0.23 ** | 0.29 ** | 0.43 ** | 0.48 ** | 0.27 ** | 0.30 ** | 0.45 ** | 0.55 ** | – |

Abbreviations: RPNH—reverse punch no hip included; RPH—reverse punch hip included; RPSM—reverse punch in motion; t—time of an event (ms); H—hand; B—body; A—acceleration in the dominant axis of motion ($g_0$); V—velocity (m/s); Ra—rotation angle (deg); S—start. Note: **. Correlation is statistically significant at the $p < 0.01$ level (2–tailed); *. Correlation is statistically significant at the $p < 0.05$ level (2–tailed).

## 4. Discussion

The primary objective of the study was to examine the variability in RP kinematic and temporal parameters due to differences in the modality of performance and to investigate their relationship. The main findings obtained in the present study were as follows:

(i)     There are significant differences in the temporal and kinematic variables of RP that arise from the modality of execution.

(ii)    Unlike kinematics, the temporal parameters show a tendency towards consistency in the more demanding modalities, which are RPH and RPSM.

(iii)   Medium and large correlations were found between the investigated temporal and kinematic variables of the body and hand.

The following considerations must be taken into account while evaluating the findings: a certain set of motor, cognitive, and perceptual abilities that are gradually enhanced with continuous training, especially in terms of processing speed and execution timing, are necessary for elite-level karate [52,53]. The applied test, which consisted of three modalities of RP, was not task- or goal-oriented but replicated typical combat time-limit situations, forcing athletes to adapt as fast as possible while executing a series of punches.

### 4.1. Differences in the Temporal and Kinematics Variables

The main finding in our study is that the execution modality causes considerable changes in measured quantities of RP. However, the results of the pairwise comparison suggest caution and additional clarification regarding RPH and RPSM. As for the seven temporal variables, only two—tHAS and tBA—were significantly different between these two modalities. On the contrary, when it comes to the kinematic variables, the difference was confirmed in all cases but HA. Indeed, HV, BA, BV, and BRa are significantly higher starting from RPNH to RPSM, although the noticeable lowest effect size was evidenced for body rotation angle and body and hand velocity between the RPH and RPSM modalities. According to certain authors, the lower body plays a major role in punch execution [1,42] and may affect preimpact hand velocity [19]. Quinzi et al. [54] observed that changing conditions of execution affects how fast the punch will be. The *gyako-tsuki* (Japanese for RP) performed from a static position reaches as high as 6.28 m/s, while dynamic execution increases the obtained values up to 6.47 m/s. In our study, the static modality RPH mean was 6.20 ± 0.70 m/s, while the highest velocity obtained in motion, RPSM, reached 6.58 ± 0.64 m/s. On the contrary, the lowest values were obtained in static modality, excluding the lower part of the body: 5.92 ± 1.64 m/s. Thus, the inclusion of the lower part of the body in RP, as well as specific test requirements (i.e., static punch execution vs. punch



execution in motion), represents the fundamental principle of RP execution and, as such, the significantly changing values of kinematic variables across the three modalities in the present study can be explained.

However, the present study provides interesting evidence regarding hand acceleration on the dominant axis. No significant difference in HA was confirmed between RPH and RPSM, and this finding must be questioned. While training without a partner is not unusual [14], a distinctive place in combat karate belongs to practicing against an opponent, with or without the use of additional equipment, adopting a combat stance. Studies have shown that with the change of performance conditions or goal orientation, the kinematics of the performed technique also change [1,28,55,56]. Different kinematic patterns can be recognized with regard to no-impact and impact-kicking actions [56]. Moreover, more experienced athletes have different performance strategies compared to less skilled ones [20,23,55]. Loturco et al. [1] determined that changing execution conditions alters the acceleration values of a punch. Tending to generate impact and choosing distance, elite athletes achieve higher acceleration in comparison to fixed distance or the intention to generate speed. Moreover, speed and self-selected distance result in higher acceleration than speed and fixed distance. Bolander et al. [28] also found that a punch performed from a long distance was significantly greater in terms of force than one from a short distance. The same also applies to arm acceleration. The dynamic sports performance environment imposes the adaptability of complex motor patterns. Athletes are forced to choose their movement strategies through a self-organization process in order to discover a proper response within these limits [42]. Apart from goal orientation, it is not negligible if the athlete is seeking an appropriate distance with regard to the opponent or target. The main point is that distance is always determined in relation to the target, whatever it may be. In the present research, the athletes were able to choose the distance themselves, but they had neither a specific goal orientation nor a target, which may explain the results. The modality of execution with a target was not employed in this study. Hence, it is not possible to offer direct evidence in this regard. Nevertheless, the absence of a significant difference in HA between RPH and RPSM might be explained in light of the presented base of knowledge. Based on that, it can be assumed that the inclusion of different types of targets (human, wooden beam, punching bag, or pad) in the regular training of elite karate competitors may be an important practice.

The fact that sports practice determines the type of motor response arising from the specific sports challenges brings another important issue regarding variability and its possible levels in a practical sense. Constantly exposed to alterations in the combat environment [1,57], karate athletes develop cognitive abilities and efficient attention processes, which allow them more time for the organization of motor behavior [58]. Elite athletes display high levels of complex neurocognitive functions, as seen in the ability to acquire complex movements [59,60]. Being able to learn implicitly with success [60], they probably develop such motor flexibility that allows them to manipulate their skills through the optimal engagement of biological capacities, thus adapting to performing tasks with rationality. For these reasons, it is important to constantly alter the training environment in terms of distance, task-goal orientation, training with or without a partner or a target, etc.

### 4.2. The Temporal Variables' Consistency in Demanding Modalities

As explained above, the inclusion of the lower part of the kinetic chain in the execution, as well as the dynamism of the RP, explains the significant difference in the temporal parameters between the RPNH and the other two modalities. The lack of significance between RPH and RPSM, however, is noteworthy. The almost complete consistency in the timeline between these modalities could be due to an ideal coordination pattern [23,59]. To be clear—apart from the sliding movement, the second and third modalities entirely shared the same execution foundation. In other words, precise inter-joint coordination between the arm and the torso occurs according to the established pattern [59] just after the athlete assumes the combat stance in the dynamic execution stage. This is indeed important

because it indicates that meeting the dynamic execution stage in punch performance will enable the replication of a similar coordination pattern, regardless of the preparatory phase. This also explains the absence of a significant difference between temporal variables in the study.

Karate requires fine control of movement in both static and dynamic situations, as well as a great ability to execute the technique as fast as possible. These skills are progressively improved with practice [52,54]. It has been determined that as expertise grows, motor performance reaches a higher level of stability and control [23,61]. In addition, coordinative patterns exhibit less variability [23] that, in relation to RP, can be proven through timeline consistency. Experience is recognized as an important factor in the structure of complex techniques that depend on multi-joint engagement [55]. The same applies to persistency in motor pattern coordination [23]. Considering the elite level of the participants in the study, it may be that persistency in the timeline is a valid parameter of technique efficiency in the overall changing combat environment simulated in the applied modalities. In other words, quantities such as kinematics are more likely to change in a predictable manner, while the time of their occurrence will remain relatively stable. This is in line with previous findings [9] that highlighted the differences in the temporal pattern of the technique, differentiated according to the achieved velocity of RP.

In practical terms, such findings suggest that dealing with a time pattern requires careful training planning, meaning that increasing the level of variability through modalities of execution may not necessarily lead to change. Technique practice in basic conditions focused on the different aspects of execution, but combined with training based on overall environment alterations, may result in proper modifications. Such an approach may result in maintaining the similarity of the timeline pattern and improving the kinematics. However, caution is suggested.

### 4.3. Correlations between the Temporal and Kinematic Variables

The conducted study showed that punch execution is most affected by the positive correlation between the following pairs of variables: BRa:BV, tBA:tBV, tBA:BA, and tHA:tHV, explaining between 30 and 57% of the variance. Loturco et al. [1] reported that 56–65% of the variation in punch acceleration can be explained on account of power and strength in the upper and lower limbs, indicating that the remaining variation is likely due to technical factors. A high level of synchronization throughout all body segments is the main characteristic of the most efficient motion pattern in throwing or striking sports involving ballistic movements [62,63]. As a result of optimal timing (including peak angular velocity of the pelvis, torso, arm, forearm, and hand) and accurate activation of the proximal to distal segment, energy transfer efficiency through the kinetic chain is maximized [63]. Thus, punch execution depends on the optimal ratio of kinematic parameters and their timely inclusion in the motion sequence.

This line of performance in the present study was identified, to some extent, through the analysis of pairs of large- and medium-correlating kinematic and temporal parameters as follows: the earlier the onset of body acceleration is evidenced, the earlier the punch will reach maximal velocity. In addition, the time of maximal body acceleration will positively influence the maximal hand acceleration, and the maximum hand velocity will occur within the optimal timeline. Moreover, higher body acceleration will occur earlier in a timeline and affect higher hand acceleration. The same applies to tHA and tHV: The earlier the maximum hand acceleration occurs in a timeline, the earlier the maximum velocity of the punch will be registered. It can be argued that the correlation evidenced between BRa:BV and BRa:tBR constitutes a mechanism of lumbopelvic control, also affecting RP execution. It has been reported that in baseball, the torso contributes up to 50% of the kinetic energy and force output over the course of a throwing motion [64]. Evidence also suggests that the rotation of the pelvis and hips affects the angular velocity of the arm movement during a punch [22]. To ensure the effective transfer of energy in a punch, it is not enough to

just attain a certain angle, the rotation of the hip must be vigorously stopped at the right point [45]. The results of our study confirmed this.

In terms of elite athletes' training practices, such findings highlight the necessity of constantly backing up the basics of karate, which are common for novices. Movement control, timed contraction, and coordination should have a distinctive place in the basic training cycle. Such a strategy creates an important foundation for upgrading the efficient technique through an advanced approach.

Finally, it is fair to say that the correlation coefficients obtained in our study do not suit the presented theory ideally. Probable causes for such evidence were the absence of goal- or task-orientation or practical guidance on how to execute punch. In other words, while performing RP consecutively, athletes were focused on continuous adaptation in a restricted time and were choosing personal strategies to reach the scoring punch. In the final analysis, punches were not differentiated according to velocity. This was done intentionally because the main goal was to reveal the characteristics of the punch in real-time situations, determining how well participants can repeat the execution sequence that results in a point-worthy action.

### 4.4. Contribution of the Study

The study provided a theoretical and practical contribution to the existing base of knowledge. Two main domains reflecting these findings are: (i) multidimensional area fusing engineering and sports issues, and (ii) improving the training process in punching techniques in combat sports.

Real-time data synchronization that combines multimodal sensors and camera fusion is not a common approach because of numerous constraining factors in data acquisition. Nevertheless, one-sided observation, from the point of view of the technical improvement of measuring instruments, does not lead to practical progress. However, unconventional engineering solutions combined with feedback from sports experts gained through application in a standardized environment result in multi-party improvements such as measurement, instrumentation, and enhancement of estimation techniques in the field of sports. It is of utmost importance that the novel solutions are reliable, easy to use, and inexpensive, offering real-time feedback in measuring, controlling, and monitoring athlete performance.

The principle of punch performance investigated in this study is shared by all combat sports. That makes the results of the study surpass the scope of karate itself in regard to the developmental phase of the punch. The significance of the impact as the most important stage of the execution cannot be denied. Nevertheless, the initiation of the punch, including both the changing curve of kinematic and temporal parameters, reveals the origins of the RP. In other words, potential deviations or preferable patterns of execution arise in the developmental phase of the technique and determine the final outcome. High-level sports achievements depend on overcoming small and hard-to-detect differences. Due to that, the accuracy and precision of the measurement devices that are insightfully applied in a proper setup are essential. Changing the structure of the punch in different modalities reveals how coaches can influence the improvement of the technique. That is accomplished only by carefully planning specific training tasks based on different goals.

### 4.5. Limitations of the Study and Future Research

It has to be stated that the present study has a few limitations. Despite the fact that the seven male participants were members of an elite team and European and world medalists, the sample can be considered limited with regard to size and gender. The absence of goal orientation and a target might have influenced the observed kinematics. Nevertheless, in light of the proposed modalities, these requirements should be considered as the next level of variability, and the proposed test should not be rejected in future research. Regarding the correlations, it would be wise to perform analysis on the set of data differentiated according to the maximum velocity or take into account only the best execution from every trial. Moreover, the final phase of the punch was not taken into account, but such analysis would

provide a more comprehensive understanding of the technique's performance. Finally, the knowledge gap regarding sensor application in combat sports stays open in many ways: comparing the placement of sensors in different locations in order to upgrade data acquisition, applying sensors with a higher operating range, combining the use of force and kinematic sensors, combining the use of sensors on the athlete and equipment, etc.

## 5. Conclusions

The results of the study confirmed that the exclusion or inclusion of body segments, as well as the dynamism of performance, affect kinematic parameters. Large and medium correlations are obtained between measured quantities. In terms of practical application, in situ data acquisition enables the understanding of the true extent of the kinematic and temporal differences of the analyzed punch. Moreover, specific tests combined with the insightful positioning of the measurement devices enable unbiased evaluation as a basis for improvements in training and competition. Such an approach shifts the training focus of elite athletes to the distinct details shared by combat sports. This leads to enhancement technique analysis and surpasses the scope of karate itself.

**Author Contributions:** Conceptualization—N.K.; methodology—N.K.; software—A.K. and A.U.; validation—A.K.; formal analysis—M.D.; investigation—A.K., A.U. and S.M.; data curation—A.K., A.U. and S.M.; writing—original draft preparation—V.V.; writing—review and editing—V.V., N.K. and A.U.; visualization—V.V.; supervision—M.D. All authors have read and agreed to the published version of the manuscript.

**Funding:** This work is sponsored in part by the Slovenian Research Agency within the research programme ICT4QoL—Information and Communications Technologies for Quality of Life (research core funding no. P2-0246) and within the bilateral project between Slovenia and Serbia titled "Sensor technologies as support systems for the detection and selection of talents in sport and monitoring the performance of athletes" (research core funding no. BI-RS/20-21-023). This paper is a part of the project "Effects of the Applied Physical Activity on the Locomotor, Metabolic, Psychosocial and Educational Status of the Population of the Republic of Serbia" (no. III47015) funded by the Ministry of Education, Science and Technological Development of the Republic of Serbia-Scientific Projects 2011–2019 Cycle.

**Institutional Review Board Statement:** The study was conducted in accordance with the postulates of the Declaration of Helsinki and was approved by the Ethics Committee of the University of Belgrade Faculty of Sport and Physical Education (02 No. 484-2).

**Informed Consent Statement:** Informed consent was obtained from the subject involved in the study.

**Data Availability Statement:** The data presented in this study is available on request from the corresponding author. The data is not publicly available due to privacy restrictions.

**Conflicts of Interest:** The authors declare that there is no conflict of interest regarding the publication of this paper.

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
