# Peer review of "Variability and the Correlation of Kinematic and Temporal Parameters in Different Modalities of the Reverse Punch Measured by Sensors"

_applsci, doi:10.3390/app131810348_

Round 1
Reviewer 1 Report
Thank you for the opportunity to review this paper. While the findings of this study are intriguing, this study has some weaknesses that need to be addressed in order to improve the quality of this paper.
First, the intorduction lacks a clear research rationale and strong research gaps may limit this study's credibility and understanding. Moreover, it would be better to have more relevant literature on Kinematic and Temporal Parameters to highlight the existing knowledge in the filed.
Second, in general, the methods and resutls are clearly explained and interpreted, and I appreciate the authors' effort into the research design and data analysis.
Third, although the empirical data presented is interesting, the discussion needs to highlight more about the contriburtin of this study to existing knowledge. Moreover, I bellive the findings can have some insightful practical implciations. I suggest the authors to restructure the discussion by adding theoretical implications and practical implications to highlight the contributions of this study to academic and practices.
Good luck.
The language is clear but needs minor editing and revision.
Author Response
Dear Editor and Reviewers,
Thank you for giving us the opportunity to submit a revised manuscript entitled Variability and the Correlation of Kinematic and temporal parameters in different modalities of the reverse punch measured by sensors for publication in the Applied Sciences special issue. We want to extend our thanks and appreciation for your time and effort, as well as the insightful comments. The authors got reviews with specific requirements and views of the quality of the study in some parts opposing each other. We tried to find a compromise and take into account the reviewers' comments to the extent that we felt it was consistent with our research. Suggestions made by the reviewers are incorporated in the revised manuscript. The changes are marked using the "Track changes" function. The additional explanations required by the reviewers are also highlighted within the manuscript and specified in detail below.

Reviewer 2 Report
As per notes attached

Author Response

(The authors gave the same response as above.)
